# Three-Dimensional Printing and Its Potential to Develop Sensors for Cancer with Improved Performance

**DOI:** 10.3390/bios12090685

**Published:** 2022-08-26

**Authors:** João B. M. Rocha Neto, Juliana Coatrini Soares, Guilherme A. Longhitano, Andrey Coatrini-Soares, Hernandes F. Carvalho, Osvaldo N. Oliveira, Marisa M. Beppu, Jorge V. L. da Silva

**Affiliations:** 1Three-Dimensional Research Group, Renato Archer Information Technology Center—CTI, Campinas 13069-901, Brazil; 2Federal University of Alagoas, Maceió 57072-900, Brazil; 3São Carlos Institute of Physics, University of São Paulo, São Carlos 13566-59, Brazil; 4School of Chemical Engineering, Department of Process and Product Development, University of Campinas, Campinas 13083-852, Brazil; 5National Institute of Biofabrication (INCT-BIOFABRIS), Campinas 13083-852, Brazil; 6Nanotechnology National Laboratory for Agriculture (LNNA), Embrapa Instrumentação, São Carlos 13560-970, Brazil; 7Institute of Biology, Department of Structural and Functional Biology, University of Campinas, Campinas 13083-86, Brazil; 8National Institute of Science and Technology for Photonics Applied to Cell Biology, University of Campinas, Campinas 13083-864, Brazil; 9School of Chemical Engineering, Department of Materials and Bioprocess Engineering, University of Campinas, Campinas 13083-852, Brazil

**Keywords:** three-dimensional printing, additive manufacturing, sensors, cancer, early diagnosis

## Abstract

Cancer is the second leading cause of death globally and early diagnosis is the best strategy to reduce mortality risk. Biosensors to detect cancer biomarkers are based on various principles of detection, including electrochemical, optical, electrical, and mechanical measurements. Despite the advances in the identification of biomarkers and the conventional 2D manufacturing processes, detection methods for cancers still require improvements in terms of selectivity and sensitivity, especially for point-of-care diagnosis. Three-dimensional printing may offer the features to produce complex geometries in the design of high-precision, low-cost sensors. Three-dimensional printing, also known as additive manufacturing, allows for the production of sensitive, user-friendly, and semi-automated sensors, whose composition, geometry, and functionality can be controlled. This paper reviews the recent use of 3D printing in biosensors for cancer diagnosis, highlighting the main advantages and advances achieved with this technology. Additionally, the challenges in 3D printing technology for the mass production of high-performance biosensors for cancer diagnosis are addressed.

## 1. Introduction

There is a continuous need to improve the methods for cancer diagnosis and decrease the high mortality associated with late diagnosis [1]. The most-used methods to detect cancer include mammography, tomography, tissue biopsies, endoscopy, and magnetic resonance imaging [2,3]. With such methods requiring expensive or invasive procedures, early diagnosis is normally hampered and the treatment may be limited or even ineffective [4]. In this scenario, the use of biosensors to detect cancer biomarkers may improve the availability and accessibility of diagnostic tools [5,6]. These include electrochemical biosensors comprising a biorecognition element (proteins, enzymes, antibodies) responsible for detecting the analyte (tumor markers, for instance) and an electrochemical transduction element [7]. The electrochemical response may be acquired with methods such as voltammetry, potentiometry, conductometry, impedance spectroscopy, and stripping techniques [8]. The detection of cancer biomarkers in liquid biopsy samples such as blood, urine, and saliva [9,10,11] is a rapid, non-invasive procedure for early diagnosis without requiring a tumor location [12]. There has been considerable progress in identifying biomarkers and fabricating biosensors, mostly using conventional 2D manufacturing processes [13,14]. Major biosensors for cancer diagnostic purposes are based on tumor markers, which indicate not just the presence of the disease but also the real-time situation of tumors. However, it is relevant to enhance the sensitivity and selectivity of the detection methods. For instance, Yin and co-workers assembled an autofluorescence-free biosensor with high analytical performance on prostate-specific antigen (PSA) detection [14] under optimal conditions. PSA is a biomarker used as an indicator for prostate cancer [15], but it is found at low concentrations in human serum, which demands biosensors with high specificity and sensitivity for early diagnosis. This can in principle be performed with biosensors made with 3D printing, which is advantageous for providing suitable fabrication procedures for high-performance biosensors. Lv and co-workers reported a 3D-printed device to develop a plasmonic-enhanced photoelectrochemical aptasensor for carcinoembryonic antigen detection, also accomplishing the purpose of detection with portable instruments [16]. Using quantum size-controlled engineering, Li and collaborators developed an efficient platform to detect HPV-16 [17], in which the device size was a key factor to adjust biosensor performance. Indeed, the size-controlled device opens a new horizon for amplification strategies, which can offer advantages in cancer diagnostics [18].

In addition to the fabrication of sensors and biosensors, 3D printing has been used in several areas such as in the production of micromachines and the development of lab-on-a-chip systems. The increasing number of publications associated with the use of 3D printing for sensors to detect cancer is illustrated in Figure 1.

Three-dimensional printing is a tool for producing semi-automated, user-friendly, low-cost, selective, and sensitive protein biomarker sensors, which can also be used to obtain final commercial products [7,18]. The principle of additive manufacturing in 3D printing has been explored to fabricate complex sensor devices with high resolution and personalized design. The sensors may be tailored to the researchers’ and clinicians’ needs, as in the detection of nucleic acids [19,20], drugs [21], proteins [22,23], trace elements [24], and neurotransmitters [25]. In comparison to traditional subtractive manufacturing, 3D printing permits to reduce the time of sensor development. Furthermore, the flexibility in sensing design and the variety of printable substrate materials allow for the development of sensors with a wide range of properties, including transparency, electrical conductivity, elasticity, and chemical and thermal resistivity [26].

In this review, we present the concepts of 3D printing and its advantages for fabricating sensors. We also provide a summary of recent 3D-printed biosensors for cancer diagnosis, highlighting the design of complex hybrid sensors achieved with 3D printing. Finally, we discuss the perspectives and challenges regarding the use of 3D printing for sensors.

## 2. Additive Manufacturing

Additive manufacturing (AM) is a fabrication process in which materials are joined layer upon layer to make parts from 3D model data [27]. It is frequently referred to as 3D printing (mainly for low-end machines) [27], and in the past it was also called rapid prototyping, rapid manufacturing, and solid freeform fabrication [28]. Since the first commercial system with stereolithography (SLA) [28,29] was proposed in 1987, AM continued to evolve with the expiration of patents and low-end machines gaining notoriety in the past decade [30,31,32]. Advantages of AM include the freedom of design to produce complex geometries, a low number of manufacturing steps, rapid production (of unique or small batches of parts), and direct-from-CAD production [33,34]. AM may be used for polymers, ceramics, metals, and composites, in the form of a filament, wire, powder, sheet, or liquid. The feedstock is processed using a laser, electron beam, ultrasound, extrusion head, print head, and light, among others [35]. AM is classified into seven categories according to the type of employed materials and processing techniques [27].

The most disseminated category is based on material extrusion through a nozzle [27], normally with desktop 3D printers (low-end machines) [35]. Examples include fused filament fabrication (FFF) and fused deposition modeling (FDM). In the vat photopolymerization category, a liquid photopolymer is maintained in a vat and cured by light. This category includes SLA, continuous liquid interface (CLIP), and direct light projection (DLP) processes. In the material jetting category, droplets of material are deposited which can be followed by curing with light, as in PolyJet and MultiJet printing (MJP). The binder jetting category uses a binder (liquid bonding agent) to join loose powder particles, which is the case in ColorJet printing (CJP). The sheet lamination category, in which material sheets are bonded to create the part, encompasses laminated object manufacturing (LOM) and selective deposition lamination (SDL). In the powder bed fusion category, thermal energy fuses or sinters a powder bed, as in selective laser melting (SLM), electron beam melting (EBM), selective laser sintering (SLS), and multi-jet fusion (MJF) techniques. The seventh category involves direct energy deposition where focused thermal energy is used to melt materials as they are deposited. Examples are laser engineered net shaping (LENS), direct metal deposition (DMD), and 3D laser cladding [27,35,36].

AM has been used from prototyping to end-use parts, in fields such as the aerospace, biomedical, automotive, educational, jewelry, architecture, and paleontology fields [34,36]. Other techniques emerged from AM, including bioprinting, a tissue engineering process explored since 2003 [37]. Bioprinting uses a bioink as deposition material [38], which is “a formulation of cells suitable for processing by an automated biofabrication technology that may also contain biologically active components and biomaterials” [39]. It consists of fabricating a three-dimensional geometry containing live cells, using an inkjet or extrusion nozzle, or a laser-assisted print head [40,41]. Inkjet bioprinting, based on material jetting AM, uses a thermal or piezoelectric actuator to deliver drops of liquid bioink selectively onto a substrate. This process permits high printing speed and resolution, but it works only with low-viscosity materials and may be damaging to the cells in the bioink owing to thermal/mechanical stresses. For extrusion bioprinting, a material-extrusion AM process, the bioink is extruded through a nozzle using pneumatic or mechanical dispensing systems. Instead of droplets, continuous beads are deposited from materials such as hydrogels, copolymers, and cell spheroids, at various viscosities and with the capability to deposit high cell densities. However, it has a low printing speed, and the high shear stresses may cause rupture in cell membranes and lower cell viability. Laser-assisted bioprinting is based on a laser-induced forward transfer with a pulsed laser beam to transfer the bioink from a donor transport support to a substrate. This process can deposit a high cell density with high cell viability but still presents a high cost and it is time-consuming for multiple cell or material arrays [40,41]. Bioprinting is used to potentially manufacture tissues and organs for transplantation or restoration, or to obtain 3D models that are more accurate than traditional cell cultures to study diseases and drug pharmacodynamics [42,43]. Bioprinting can also yield three-dimensional biosensors for disease diagnostics and biosecurity applications [44,45]. Figure 2 presents schematic models for some AM techniques.

## 3. Three-Dimensionally Printed Sensors for Cancer Diagnosis

Three-dimensional printing is convenient for fabricating biosensors due to its low cost, as it can be performed with photolithography to reach high-resolution devices that may be adapted. For instance, 3D-printed electrodes provide versatility in the geometry and functionality of electrochemical sensors. The electrochemical transduction elements normally contain noble metals (gold, silver, platinum), carbon, or organic and inorganic semiconductors [46], and 3D printing allows for controlling the sensor surface properties to maximize tumor cell adhesion [47,48]. Hence, with 3D printing, it is possible to improve the selectivity and sensitivity of biosensing platforms. Three-dimensional printing may be customized depending on the type of material used. The most common techniques to produce sensors are: (1) fused deposition modeling (FDM), which combines non-conductive filaments (polylactic acid, acrylonitrile butadiene styrene) with conductive carbonaceous materials (graphite, graphene, carbon nanofibers, carbon nanotubes, and carbon black), and (2) selective laser melting (SLM), which uses metal powder (iron, steel, and aluminum) to fabricate electrodes [49]. Recent studies on 3D printing for sensors related to cancer diagnosis are described below.

An electrochemical sensor was obtained with 3D printing for detecting the response of cells and tissues for three colon cancers on cell culture plates via amperometry [50]. This was made possible with a sensor architecture to measure the region under test, eliminating the need for biopsies as is commonly performed with conventional diagnostic methods. The chip is composed of polydimethylsiloxane (PDMS) cast from a source fabricated by 3D printing, in which the electrical contact is enabled by conductive PDMS containing 60 wt% graphite powder. A 3D-printed flow system with screen-printed electrodes was used to detect hepatic oval cells that over-express the OV6 marker on their membrane [51]. Hepatic oval cells are found in a small number in the liver; thus, capturing them is challenging for immunosensors. The biosensors were made from carbon nanotubes and chitosan films on which oval cell marker antibodies were immobilized, thus enhancing the specificity toward the biomarker. The sensitivity was enhanced because the 3D-printed flow cell improved the capture of target cells in the sample. However, the performance of the above devices may be impaired due to the clogging of microfluidic channels by cells, which may influence the sensitivity of biosensors.

Electrochemical sensors can also be combined with acoustic sensors, using a surface-sensitive microbalance based on acoustic wave propagation along a quartz crystal. Damiati and co-workers described an efficient acoustic and hybrid 3D-printed electrochemical biosensor for real-time detection of liver cancer cells (HepG2) [52]. The biosensors explore the interaction between recombinant proteins and the tumor biomarker CD133, which is found at high concentrations on liver cancer cells. Detection of tumor markers, which are often proteins produced by cells at higher amounts in response to cancer [53], can help diagnose cancer at an early stage, guide treatment, and determine prognosis [54]. A ceramic substrate with noble metals was used for the detection element and 3D-printed capillary channels to guide and restrict the clinical sample. Cyclic voltammetry (CV) measurements confirmed the efficiency of the sensors and may have applications in the clinic and in drug screening studies. Another electrochemical biosensor made with 3D printing detected an alkaline phosphatase biomarker from three different colon cancer cell lines in a cell culture plate [50]. This approach opens the way for direct and non-invasive diagnostics on a layer of exposed cells for in vivo and in vitro applications. The 3D chip used was made of PDMS fused to a master polymer manufactured by 3D printing. Electrical contact was established with conductive PDMS containing 60% by weight of graphite powder. A stable voltammetric signature was observed, with a significant amperometric response to the enzyme.

Electrochemical luminescence has also been exploited to detect cancer biomarkers [55,56], with the transducer effect being based on electrochemical reactions in solutions where luminescent species produce an excited state that emits light through relaxation to a lower-level state. This principle was employed in a bipolar electrode system for electrochemiluminescence detection of human breast cancer cells (MCF-7) [55]. Three-dimensional printing was used to build microchannels capable of minimizing the required amounts of sample, improving detection performance. Since the diagnosis of breast cancer requires scanning the whole volume of the breast, a 3D biosensing array with 24 electrodes and different geometries was employed to produce a volumetric image of capacitance data of the breast in real-time [57]. Imaging was effective because the relative permittivity of cancerous mass is higher than in normal breast tissue. This method is a fast, non-radiation alternative for breast cancer screening, which makes it remarkable in terms of functionality and performance compared to others that do not use 3D printing. Another electrochemiluminescent immunosensor detected multi-tumor cancer biomarker proteins in serum samples, with a good correlation with results from the conventional single-protein ELISAs for six serum samples from prostate cancer patients [56]. Devices produced with fused deposition modeling (FDM) and stereolithographic 3D printers were used to detect three proteins with detection limits from 0.3 to 0.5 pg/mL [58]. Detection was carried out with electrochemiluminescence (ECL) in an open channel with integrated sensor elements printed on disposable screens. A 3D-printed prototype was controlled by a closed microprocessor microfluidic ECL immunoarray with reagent reservoirs, micro-pumps and transparent plastic detection chamber with printed nanowells for ECL emission [59].

From a more technological perspective, efforts are being made to produce lightweight, miniaturized devices for high-throughput point-of-care diagnostics, including data transmission. The availability of desktop 3D printers and printing options has turned 3D printing into a tool to develop low-cost, high-resolution complex immunosensors, tailored to users’ needs. Leveraging the increasing number of connected devices, researchers developed an efficient mobile-phone-based electrochemical biosensor for point-of-care applications [60]. These sensors were able to monitor, with a low limit of detection, the concentration of a biomarker for tracking lung infections in cystic fibrosis patients via electrochemical measurements. A custom smartphone multi-view app combined with an optical 3D sensor was capable of monitoring the sensing parameters and measuring the concentrations of a human cancer biomarker [61]. In another study, a magnetic tracked 3D sensor allowed for 3D image acquisition with an endoscopic probe to detect vessel involvement in pancreatic tumors [62]. The versatility of 3D printing for electrode fabrication makes it possible to apply a wide variety of materials that can be used in electrode functionalization. Because it confers homogeneity on molecular architecture [63], 3D printing can improve analytical characteristics such as selectivity and sensitivity. An efficient strategy to capture and detect rare circulating tumor cells (CTCs) in the blood of cancer patients was developed using a polymeric substrate containing gold modified with benzoboric acid and a regular 3D surface array [64]. The 3D surface exhibited a 3.8-times higher capture efficiency, as compared to a smooth substrate. This is promising for early diagnosis of cancer due to the high sensitivity, low cost, and recovery of isolated CTCs. In another example [65], 3D printing was used to construct scaffolds with precise macroporous structures to monitor the behavior of mammalian cells, with which one can investigate disease progression and drug efficacy. Detection was made using surface-enhanced Raman scattering spectroscopy (SERS) with substrates obtained on a 3D-printed framework of a plasmonic hydrogel. Early diagnosis of cancer can be carried out by detecting biomarkers that are overexpressed in body fluids and associated with different types of cancer [66].

Biofabrication based on 3D printing permits the unique capability of manufacturing tumor models [67], which can be exploited for diagnosis. Indeed, it has been tested for different drugs to eliminate risks and improve treatments. Chiadò and co-workers reported a polymer-based 3D device for detecting protein biomarkers related to angiogenesis, which are key factors to monitor the metastatic behavior of tumors [68]. The 3D-printed device was developed in a single-step printing process using the stereolithography method, which was crucial to obtain a polymeric chip with intrinsic tuning design and functionality.

### Advantages of 3D Printing in the Fabrication of Biosensors

Analytical devices have been developed with 3D printing for the early detection of diseases, including cancer, diabetes, and COVID-19 [69,70,71], in many cases allowing for portability and application in hospitals and clinics. Three-dimensional printing is advantageous due to the possible mass production of devices with enhanced physical stability, and control over the fabrication process, from the type of paint to functionalization and electrode geometry [72,73,74]. This has allowed biosensors to be produced with matrices made of polymers, proteins, and genetic material. Manufacture can be carried out with a single platform with the device components printed on a single device, including the active layer [72]. Alternatively, it can be performed with a multimodal platform, in which physical and chemical components can be incorporated into commercial devices such as electrodes [75,76] and light-addressable potentiometric sensors (LAPS) [77]. Since 3D-printed biosensors are normally based on the same transduction principles as other biosensors, they also incorporate similar components. For instance, 3D-printed biosensors may have antibodies, nucleic acids (DNA/RNA), cells, and aptamers as recognition elements to detect analytes [78]. This is made possible with the versatility of 3D-printing techniques, as the conducting ink can be chemically modified and the electrode architecture can be tailored to obtain high sensitivity [79,80], low material waste [81], and fast fabrication [82]. The sensitivity and selectivity of 3D biosensors can be enhanced by modifying the electrode surface, specific for each type of analyte. For example, a thin layer of gold can be sprayed on electrode carbon surfaces to reduce surface resistivity. Furthermore, 3D biosensors have increased surface/volume ratios, with less variability in measurements which contributes to a higher reproducibility [83]. The stability of the biorecognition layer in 3D biosensors is normally higher as the layer is less exposed to the environment than in 2D biosensor platforms. For this reason, non-specific adsorption is also less likely in 3D biosensors [83].

In a comparison with current techniques to fabricate biosensors, e.g., photolithography, screen printing, or roll-to-roll, 3D printing is outstanding, because it requires a lower number of human-made operations in a production process with fewer errors, high accuracy, and repeatability [84,85]. Three-dimensionally printed biosensors are also robust against physical damage generated both in the electrode printing process and during the etching process. Such robustness permits the biosensors to be employed under varied detection conditions with changes in pH, temperature, and mechanical stress. Furthermore, 3D printing allows for the use of a larger number of materials than photolithography, and the resulting biosensors possess higher durability and mechanical resistance than those produced with screen printing or the roll-to-roll technique. Customized 3D-printed biosensors can be obtained with techniques such as digital light processing (DLP), and then be tailored for specific analytes [85,86].

## 4. Current Challenges and Perspectives

In the last section, we emphasized the strengths and advantages of 3D manufacturing for biosensors, but there are still major limitations that may prevent the dissemination of 3D-printed biosensors in the market [85]. These limitations include the long printing times and the impossibility of printing multiple materials on the same 3D printer [87]. Nanoparticles rejected in 3D printing can also be harmful to the environment [88]. Since biosensor fabrication requires different materials, the instrumentation for 3D printing should be versatile enough to print materials with different physical and chemical characteristics within a short time. Indeed, immunosensors and genosensors usually contain various materials [89,90,91] since their molecular architecture comprises a matrix to help preserve the activity of the biomolecules in the active layer. Most of the described studies indicate that biosensors produced with 3D printing are affordable due to the low cost of the technique and 3D printers [66]. Moreover, 3D technology makes it possible to quickly and optimally immobilize proteins and other analytes on the same platform [92]. These factors make 3D printing suitable for detecting biomarkers of all types of cancer and other diseases, capable of multiplexing protein biomarkers that could be correlated with different abnormalities and different types of cancer.

Developing new 3D electrodes is crucial for improving analytical parameters such as sensitivity and selectivity, and this can be exploited with a myriad of technologies, materials, and sensor geometry employing 3D printing. Different types of substrate can be produced, including rigid, flat substrates from ceramics or silica, and flexible substrates from plastics and paper. Three-dimensionally printed electrodes are low-cost, but the resulting biosensors have low production efficiency and processing difficulties [93]. One of the consequences is the limited reproducibility, with dispersion in the sensing parameters above 10% for nominally identical sensing units [94]. Moreover, analytical parameters such as sensitivity may be inferior to those of biosensors built using conventional techniques. Efforts to improve these parameters encompass optimization of ink type, curing type, ink distribution on the substrate, and electrode geometry [95,96]. For instance, Cagnani and colleagues [97] manufactured screen-printed carbon electrodes modified with carbon black and enzymatic inks to detect dopamine, with high reproducibility and low limit of detection (0.09 μmol/L). As for the electrode geometry, it is especially relevant in impedance spectroscopy since the electrical signal (capacitance) depends on the width of the capacitor distance on the working electrode [98,99]. Interdigitated electrodes (IDEs) built from parallel plate capacitors are normally used to maximize sensitivity and selectivity [100,101]. With 3D printing, one cannot obtain the same resolution for interdigitated electrodes as existing techniques such as photolithography, with which biosensors have been made for cancer diagnosis and food quality control [102,103,104]. In fact, this limited resolution is highlighted in recent works using 3D-printed IDEs [105,106].

Regarding biosensors, 3D printing has been mostly used to replace techniques such as photolithography, owing to its low cost and short manufacturing times. It has also been employed to obtain functionalized intelligent materials where the active layer is printed with conducting inks modified with proteins, antibodies, and genetic material. The production of electrodes with biocompatible inks based on carbon and noble metal materials is still a major challenge in 3D printing. Though 3D printing on carbon materials is less costly than photolithography, it is limited owing to the smaller resolution. Indeed, with current methods, 3D printing is not suitable for some applications with large-area electrodes, especially in interdigitated electrodes. One hopes that recent advances in nanotechnology may allow conductive inks to be printed on electrodes with higher resolution. We expect this trend to continue, as the demand for low-cost tests that are manufactured quickly is increasing with the continuation of the COVID-19 pandemic, in addition to the aims of developing personalized medicine. Developments in 3D printing have a multidisciplinary nature through the convergence of materials engineering, design, physics, chemistry, and biology. The synergy in contributions from these various areas permits the creation of new analytical devices with high resolution, which may be portable to facilitate prognosis/diagnosis of diseases in places of difficult access (such as space stations, isolated communities) and countries with poor access to public health services.

## Figures and Tables

**Figure 1 biosensors-12-00685-f001:**
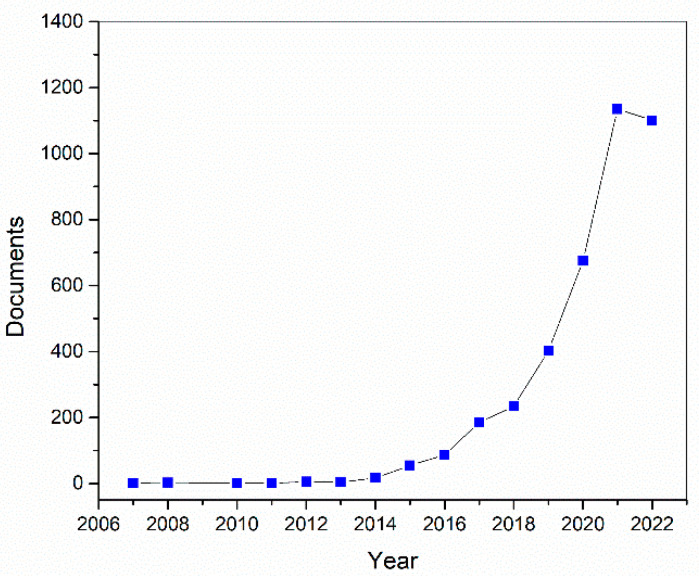
Number of publications retrieved with a search with keywords “cancer”, “sensor”, and “3D-printing” from 2007 to 2022. Results were generated using Scopus^®^ report generation tool on 6 August 2022.

**Figure 2 biosensors-12-00685-f002:**
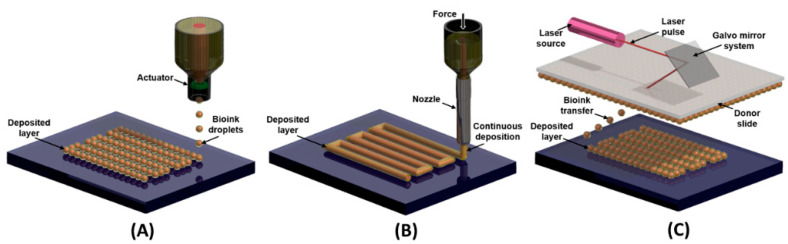
Schematic models for (**A**) inkjet, (**B**) extrusion, and (**C**) laser-assisted bioprinting techniques.

## Data Availability

Not applicable.

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
