# Peer review of "Three-Dimensional Printing and Its Potential to Develop Sensors for Cancer with Improved Performance"

_biosensors, 2022, doi:10.3390/bios12090685_

Round 1

Reviewer 1 Report

This is a very useful review of applications of 3D printing technologies in biosensing and particularly in cancer diagnostics.

The review is a bit too short and tends to describe 3D-printed biosensors in general. The enhancement of sensitivity and selectivity of 3D-printed biosensors was mentioned in the text several times without detailed explanation. One of the obvious advantages of 3D-printed sensors is combining of the fabrication of a biosensing layer and sample delivery system in one technological process. From the examples given, it was understood that the enhancement of the sensitivity could be related to the increase in the sensing surface area. 3D-printed surfaces for SERS-based sensors are also promising. Those are only two examples of the sensitivity enhancement given in the review. The enhancement of the selectivity in 3D-printed biosensors is not obvious and needs further explanation. Also, the use of microfluidic for delivery of cancer cells samples could be problematic because of the channels’ clogging; as a result such 3D-printed sensors could be used only once and therefore not cost-efficient.

English could be improved; check for example lines 51, 55, 92. 

The review could be improved after addressing the above comments within minor revision.   

Reviewer 2 Report

This manuscript summarized recent advances on 3D printing and its potentials for the development of biosensors with cancer diagnosis. Rather than being exhaustive, this review focuses on selected examples to illustrate novel concepts and promising applications. Meanwhile, the major advantages and advances achieving with this technology were highlighted. Also addressed are the challenges in 3D printing technology for the mass production of high-performance biosensor for cancer diagnosis. Further, promising applications in bioanalysis are considered and discussed.

Specific comments:

1.     Note that the review should be above all critical and interpretative rather than enumerative and that the analytical figures of merit should be listed in comprehensive Tables rather than discussing them in the main text, except in those cases where the paper shows a significant improvement.

2.     As a critical review, authors selected some examples to illustrate novel concepts and promising applications on isothermal amplifications. However, some critical comments should be described and explained in the main text. Maybe, it is better if authors can further discuss and remark these selected examples in the main text.

3.     There has been considerable progress in the identification of biomarkers and fabrication of biosensors, mostly using conventional 2D manufacturing processes. However, for various cancer diagnostics procedures one needs to enhance both the sensitivity and selectivity of the detection methods. Please provide the corresponding literatures for these descriptions (e.g., CRISPR-Cas12a-derived photoelectrochemical biosensor for point-of-care Diagnosis of nucleic acid; Size-controlled engineering photoelectrochemical biosensor for human papillomavirus-16 based on CRISPR-Cas12a-induced disassembly of Z-scheme heterojunctions; Persistent luminescence nanorods-based autofluorescence-free biosensor for prostate-specific antigen detection; Ultrasensitive fluorometric biosensor based on Ti3C2 MXenes with Hg2+-triggered exonuclease III-assisted recycling amplification).

4.     This can in principle be done with biosensors made with 3D printing, which is advantageous for providing suitable fabrication procedures for high-performance biosensors. Recent works on 3D printing (e.g., Plasmonic enhanced photoelectrochemical aptasensor with D-A F8BT/g-C3N4 heterojunction and AuNPs on a 3D-printed device. Sensors and Actuators B: Chemical 2020, 310, 127874) should be mentioned for this description. In addition, literatures on 3D printing devices for the biosensors are insufficient.

5.     In the "Current challenges and perspectives" section, authors summarized the advantages and disadvantages of 3D printing and its potentials for the development of biosensors with cancer diagnosis. Actually, authors should remark the future application and application perspectives.

6.     References: Please carefully check all references, and some irrelative references should be removed. In summary, too relative literatures on this topic are missing, and they should be cited in this review. Meanwhile, some old literatures might be updated.
